# Effects of Selective and Mixed-Action Kappa and Delta Opioid Receptor Agonists on Pain-Related Behavioral Depression in Mice

**DOI:** 10.3390/molecules29143331

**Published:** 2024-07-16

**Authors:** S. Stevens Negus, Celsey M. St. Onge, Young K. Lee, Mengchu Li, Kenner C. Rice, Yan Zhang

**Affiliations:** 1Department of Pharmacology and Toxicology, School of Medicine, Virginia Commonwealth University, Richmond, VA 23298, USA; youngkwang.lee@vcuhealth.org; 2Department of Medicinal Chemistry, School of Pharmacy, Virginia Commonwealth University, Richmond, VA 23298, USA; cstonge@scripps.edu (C.M.S.O.); mengchu9102@gmail.com (M.L.); yzhang2@vcu.edu (Y.Z.); 3Drug Design and Synthesis Section, Molecular Targets and Medications Discovery Branch, Intramural Research Program, National Institute on Drug Abuse and the National Institute on Alcohol Abuse and Alcoholism, National Institutes of Health, Department of Health and Human Services, 9800 Medical Center Drive, Bethesda, MD 20892, USA; kennerr@nida.nih.gov

**Keywords:** kappa opioid receptor, delta opioid receptor, nalfurafine, SNC80, hydrocodone, aprepitant, pain-depressed behavior

## Abstract

We recently developed a series of nalfurafine analogs (TK10, TK33, and TK35) that may serve as non-addictive candidate analgesics. These compounds are mixed-action agonists at the kappa and delta opioid receptors (KOR and DOR, respectively) and produce antinociception in a mouse warm-water tail-immersion test while failing to produce typical mu opioid receptor (MOR)-mediated side effects. The warm-water tail-immersion test is an assay of pain-stimulated behavior vulnerable to false-positive analgesic-like effects by drugs that produce motor impairment. Accordingly, this study evaluated TK10, TK33, and TK35 in a recently validated assay of pain-related behavioral depression in mice that are less vulnerable to false-positive effects. For comparison, we also evaluated the effects of the MOR agonist/analgesic hydrocodone (positive control), the neurokinin 1 receptor (NK1R) antagonist aprepitant (negative control), nalfurafine as a selective KOR agonist, SNC80 as a selective DOR agonist, and a nalfurafine/SNC80 mixture. Intraperitoneal injection of dilute lactic acid (IP lactic acid) served as a noxious stimulus to depress vertical and horizontal locomotor activity in male and female ICR mice. IP lactic acid-induced locomotor depression was alleviated by hydrocodone but not by aprepitant, nalfurafine, SNC80, the nalfurafine/SNC80 mixture, or the KOR/DOR agonists. These results suggest that caution is warranted in advancing mixed-action KOR/DOR agonists as candidate analgesics.

## 1. Introduction

The ongoing crisis of opioid addiction and overdose deaths has stimulated efforts to discover novel pain-treatment drugs that retain the effectiveness of mu opioid receptor (MOR) agonist analgesics like morphine but produce fewer side effects in general and have lower abuse potential in particular [1]. We recently reported that these goals might be met by the novel opioid NMF (referred to in this article as TK10 (NMF)). This compound functions as a mixed-action agonist at both kappa and delta opioid receptors (KOR, DOR) and is an analog of the KOR agonist nalfurafine (Figure 1) [2]. In a mouse warm-water tail-immersion assay, NMF produced potent, effective, and long-acting antinociception that was attenuated by KOR and DOR antagonists administered individually (norbinaltorphimine and naltrindole, respectively), completely blocked by the co-administration of the KOR and DOR antagonist, but unaffected by pretreatment a MOR antagonist (ß-funaltrexamine). Moreover, unlike morphine, NMF was not self-administered by rats, and in mice, it did not stimulate locomotor activity, did not produce morphine-like signs of naloxone-precipitated withdrawal after repeated dosing, and did not depress measures of respiration [2].

The warm-water tail-immersion test of antinociception used in the initial NMF studies can be classified as an assay of “pain-stimulated behavior,” in which warm water stimulated a behavioral response (in this case, the reflexive behavior of tail-withdrawal) [3,4,5]. Assays of pain-stimulated behavior can be useful during initial drug evaluation to assess pharmacological parameters such as potency and time course. However, these procedures are also vulnerable to false-positive effects by drugs that reduce the target behavior by producing motor impairment rather than analgesia. To address this vulnerability, we and others have developed assays of “pain-depressed behavior” [3,6,7,8,9,10,11,12]. In these assays, the noxious stimulus decreases expression of the target behavior (e.g., reduction in locomotion, feeding, nesting, or operant responding), and candidate analgesics are evaluated for the effectiveness in increasing expression of the pain-depressed behavior back to baseline levels. These procedures have improved preclinical-to-clinical translational validity because they retain sensitivity to clinically effective analgesics but eliminate false-positive results with drugs that produce motor impairment and only exacerbate pain-related behavioral depression. These procedures also have high face validity because pain-related behavioral depression is a cardinal sign of clinical pain (e.g., as evaluated using the Brief Pain Inventory of pain interference), and alleviation of pain-related behavioral depression is an important goal of pain treatment [13,14].

Accordingly, the major goal of the present study was to evaluate the effects of the mixed-action KOR/DOR agonist TK10 (NMF) and two other structurally related KOR/DOR agonists (TK33, TK35; see Figure 1) [15] in a recently validated assay of pain-depressed behavior in mice [16,17]. Effects of these mixed-action compounds were compared to effects produced by the selective KOR agonist nalfurafine [18,19] and selective DOR agonist SNC80 [20,21], both of which have also been found to produce antinociceptive effects in preclinical assays of pain-stimulated behavior [22,23,24,25,26,27]. Lastly, to provide additional assay validation, effects were also determined for (a) hydrocodone as another positive-control MOR-agonist analgesic [28] and (b) the neurokinin 1 receptor (NK1R) antagonist aprepitant as a negative-control non-analgesic drug [29]. Aprepitant and other NK1R antagonists were initially developed as candidate analgesics based on the role of substance P in nociceptive transmission. However, while aprepitant and other NK1R antagonists produced antinociception in various preclinical assays of pain-stimulated behavior [30,31,32], they ultimately proved to be ineffective as analgesics. Hence, NK1R antagonists represent one example of failed preclinical-to-clinical translation [30,31,32,33,34,35].

## 2. Results

Eight different drugs or drug mixtures were evaluated in this study over a period of approximately 6 months, and each drug was examined in two different cohorts of mice to test the effects of the drug administered both (1) alone and (2) as a pretreatment to IP 0.56% lactic acid. One-way ANOVA found no significant differences in the effects of the vehicle alone for either crosses (F(7,88) = 1.81; *p* = 0.0946) or movement counts (F(7,88) = 0.70; *p* = 0.6681) across successive cohorts of mice, suggesting that baseline locomotor behavior was relatively stable across cohorts. Additionally, Table 1 shows that vehicle + IP lactic acid produced significantly lower levels of both crosses and movement counts than vehicle alone in cohorts used to test each drug, suggesting that IP lactic acid-induced behavioral depression was also relatively stable across cohorts. Each drug was evaluated for its effectiveness in alleviating IP lactic acid-induced depression of crosses and movement.

Figure 2 shows antinociceptive effects by a positive-control drug (the opioid analgesic hydrocodone) but not by an active negative control (the NK1 receptor antagonist aprepitant). One-way ANOVA results for these and all other treatments administered alone or as a pretreatment to IP lactic acid are shown in Table 2. When administered alone, hydrocodone displayed an inverted-U-shaped dose-effect curve for crosses, with a significant increase at 10 mg/kg and a significant decrease at a high dose of 32 mg/kg. When administered as a pretreatment to IP lactic acid, hydrocodone produced antinociception, indicated by a significant alleviation of IP lactic acid-induced depression of both crosses (5.6 mg/kg) and movement (5.6–18 mg/kg). Aprepitant, by contrast, only decreased crosses (3.2–10 mg/kg) and movement (32 mg/kg) when it was administered alone, and it failed to alleviate IP lactic acid-induced depression of either crosses or movement.

Figure 3 shows the effects of the KOR agonist nalfurafine, the DOR agonist SNC80, and a 10:1 mixture of SNC80/nalfurafine. Nalfurafine administered alone dose-dependently decreased both crosses and movement, and it failed to alleviate IP lactic acid-induced depression of either endpoint. SNC80 alone had no effect on crosses or movement at low doses of 0.1–0.32 mg/kg, but it significantly increased crosses at intermediate doses of 1–10 mg/kg, and a high dose of 32 mg/kg significantly decreased movement; however, like nalfurafine, no dose of SNC80 alleviated IP lactic acid-induced behavioral depression. Nalfurafine alone was approximately 10-fold more potent than SNC80 alone to significantly alter locomotion (i.e., nalfurafine significantly decreased crosses and movement at doses of 0.1 and 0.032 mg/kg, respectively, whereas SNC80 significantly increased crosses at 1.0 mg/kg). Accordingly, we tested a 10:1 mixture of SNC80 and nalfurafine. Similar to nalfurafine alone, the 10:1 SNC80/nalfurafine mixture only decreased locomotor endpoints when it was administered alone, and no dose of the mixture alleviated IP lactic acid-induced behavioral depression.

Figure 4 shows the effects of the mixed-action KOR/DOR agonists TK10 (NMF), TK33, and TK35. All three compounds produced dose-dependent decreases in crosses and movement when they were administered alone. TK10 (NMF) and TK33 were approximately equipotent, with significant decreases in crosses at doses ≥ 0.1 and in movement at doses ≥ 0.032 mg/kg. TK35 alone produced qualitatively similar dose-dependent decreases in crosses and movement but was approximately 3-fold more potent than TK10 (NMF) and TK33. None of the compounds significantly alleviated IP lactic acid-induced depression of either crosses or movement.

When data were segregated by sex, there was neither a Sex × Dose interaction nor a main effect of sex for most (19/32) analyses (summary results in Table 2; detailed results in Appendix A). A significant Sex × Dose interaction was observed in only four instances (aprepitant alone for crosses, SNC80 + IP lactic acid for both crosses and movement, and TK35 + IP lactic acid for movement), and data segregated by sex for each of these experiments are shown in Appendix A. Notably, post hoc analyses indicated only isolated sex differences at individual doses, and there was no evidence for systematic effects of sex on either potency or maximal drug effects. Lastly, there was a main effect of sex in nine analyses; all five analyses showing a main effect of sex on crosses indicated higher overall scores in males, whereas all four analyses showing a main effect of Sex on movement indicated higher overall scores in females. 

## 3. Discussion

This study compared the effects of selective and mixed-action KOR and DOR agonists to the effects of a positive-control analgesic (hydrocodone) and a negative-control non-analgesic (aprepitant) in a recently validated assay of pain-related behavioral depression in mice. There were three main findings. First, in agreement with previous studies with positive and negative controls in this procedure, the MOR agonist analgesic hydrocodone significantly alleviated IP lactic acid-induced behavioral depression, whereas the non-analgesic NK1R antagonist aprepitant did not. Second, the selective KOR agonist nalfurafine and DOR agonist SNC80 also failed to produce an antinociceptive relief of IP lactic acid-induced behavioral depression. Lastly, neither the 10:1 SNC80/nalfurafine mixture nor the three mixed-action KOR/DOR agonists were effective in alleviating IP lactic acid-induced behavioral depression. Taken together, these results provide additional validation of this procedure’s utility as a preclinical tool to distinguish between clinically effective analgesics and non-analgesics. Additionally, the present results do not support the clinical potential of selective KOR agonists, DOR agonists, or mixed-action KOR/DOR agonists as promising candidate analgesics for relief of pain-depressed behavior.

### 3.1. Effects of Hydrocodone and Aprepitant

The present results agree with previous findings that clinically effective MOR agonist analgesics are effective in alleviating IP lactic acid-induced behavioral depression of both crosses and movement in this procedure, whereas many non-analgesic drugs are not [16,17]. MOR agonists can produce both analgesic effects (which increase IP lactic acid-depressed behavior) and motor disruptive effects (which can either increase or decrease behavior and obscure analgesia-associated increases in behavior). As a result, MOR agonists with relatively high intrinsic efficacy to activate MOR receptors often produce inverted-U-shaped dose-effect curves in this procedure, with analgesic effects predominating at intermediate doses and motor effects predominating at higher doses. Hydrocodone produced this profile of effects here. 

In contrast, aprepitant is representative of the class of NK1R antagonists initially developed as candidate analgesics based on evidence that substance P acting at NK1R plays an important role in the transmission of nociceptive information from primary nociceptors in the periphery to secondary nociceptors in the spinal cord [33,36]. In support of their potential utility as analgesics, aprepitant and some other NK1R antagonists produced antinociceptive effects in preclinical assays of pain-stimulated behavior [30,31,32,34]; however, these drugs ultimately failed to produce effective pain relief in humans [33,36]. As a result, NK1R antagonists represent a prominent example of failed translation of preclinical results from assays of pain-stimulated behavior. The present results are consistent with other evidence to suggest that preclinical assays of pain-depressed behavior may enable improved preclinical-to-clinical translation. 

### 3.2. Effects of Nalfurafine and SNC80 Administered Alone or in Combination

The KOR agonist nalfurafine is another example of failed translation from preclinical assays of pain-stimulated behavior. Like aprepitant and NK1R antagonists, nalfurafine is representative of the many KOR agonists that have been evaluated as candidate analgesics, and like many other KOR agonists, nalfurafine produced significant antinociception in mice, rats, and nonhuman primates tested in assays of pain-stimulated behavior using models of nociceptive, inflammatory, and neuropathic pain [22,23,24,37]. However, nalfurafine and other KOR agonists that are distributed to the central nervous system produce dose-limiting side effects that undermine any utility they might have for pain treatment [38]. The ineffectiveness of nalfurafine to alleviate IP lactic acid-depressed behavior in the present study is consistent with both its clinical ineffectiveness as an analgesic and with the absence of antinociception by nalfurafine and other selective and centrally acting KOR agonists in this and other assays of pain-depressed behavior [18,39,40]. 

SNC80 is one of several DOR agonists that has produced antinociception in at least some preclinical assays of pain-stimulated behavior, but these effects may again represent false-positive results that do not translate to clinically effective analgesia [25,26,27,41,42,43,44]. Although SNC80 itself has not advanced to clinical trials, three other DOR agonists have advanced as far as Phase II trials [45]. Two of those trials were terminated due to the lack of analgesic effectiveness. A third Phase II trial was completed in 2013, but more than 10 years later, results still have not been published or posted to clinicaltrials.gov (NCT01291901). The failure of SNC80 to alleviate IP lactic acid-induced behavioral depression in the present study is consistent with the apparent failure of these other DOR agonists to produce clinically effective analgesia in clinical trials. Results with SNC80 also illustrate another important feature of the present assay of pain-depressed locomotor behavior. When SNC80 was administered alone, intermediate doses significantly increased crosses and trended toward an increase in movement. This evidence for locomotor stimulation is consistent with other studies that have reported SNC80-induced locomotor stimulation in mice [46,47,48]; however, despite these locomotor stimulant effects when SNC80 was administered alone, it failed to increase locomotion in the presence of IP lactic acid. This finding agrees with other data reported previously to suggest that locomotor stimulant effects of drugs administered alone are not sufficient to predict or explain the alleviation of IP lactic acid-induced behavioral depression [16]. Rather, drug-induced relief of IP lactic acid-induced behavioral depression appears to reflect analgesia rather than general stimulation of locomotor behavior.

The lack of antinociceptive effectiveness by an SNC80/nalfurafine mixture is consistent with the lack of antinociception by the constituent drugs. The 10:1 SNC80/nalfurafine mixture tested here produced a profile of effects similar to nalfurafine alone in that it produced only dose-dependent depression of crosses and movement when administered alone, suggesting that the KOR effects of nalfurafine may have predominated and blocked the motor-stimulant effects of SNC80. Other mixtures with higher SNC80 proportions might produce more SNC80-like effects, but the present results suggest that other mixtures with a higher contribution of DOR-mediated effects would also be unlikely to produce antinociception. 

### 3.3. Effects of Novel Mixed-Action Kappa Opioid Receptor/Delta Opioid Receptor Agonists

TK10 (NMF), TK33, and TK35 are nalfurafine analogs that bind with similar affinities to KOR, DOR, and MOR [2,15]. However, in vitro functional assays have indicated that these compounds function as full, high-efficacy KOR and DOR agonists while having relatively low MOR efficacy, and in mice, all three compounds produced antinociception in a warm-water tail-immersion test mediated by KOR and DOR but not MOR. Additionally, the antinociception by these compounds was not accompanied by MOR-typical side effects, including abuse potential in a rat drug self-administration procedure, respiratory depression in mice, or morphine-like abstinence signs in mice after a regimen of chronic treatment followed by naloxone administration. These results were interpreted to suggest that TK10 (NMF), TK33, and TK35 might serve as effective and safer alternatives to MOR agonists for clinical treatment of pain [2,15]. 

However, in the present study, antinociception did not extend to an assay of pain-depressed behavior in mice in which clinically effective opioid and nonsteroidal anti-inflammatory drug analgesics are effective. Moreover, TK10 (NMF), TK33, and TK35 administered alone in this study decreased locomotor endpoints with a potency similar to that of their antinociceptive effects in the warm-water tail-immersion assay [2,15]. This raises the possibility that apparent antinociception by these compounds in the warm-water tail-immersion assay may have resulted from motor impairment rather than from reduced pain sensitivity. Additionally, the absence of locomotor stimulation by any of these compounds suggests that, as with the 10:1 SNC80/nalfurafine mixture, the KOR agonist effects may have predominated over DOR agonist effects. Overall, the present results do not support the hypothesis that these mixed-action KOR/DOR agonists will function as effective analgesics in humans. As a caveat to this conclusion, it should be noted that all compounds in this study were administered by an SC route of administration. Although investigation of drug effects using other routes of administration was beyond the scope of the present study, such other routes of administration may produce different pharmacokinetic profiles and could yield different behavioral results. 

### 3.4. Sex as a Determinant of Test-Drug Effects

The main goal of this study was to evaluate dose as a determinant of effects produced by drugs administered alone or as a pretreatment to IP 0.56% acid; however, each drug was tested in equal numbers of female and male mice, and results permitted preliminary analysis of sex as a determinant of drug effects. As in our previous studies with this procedure [16,17], sex was not a reliable or systematic predictor of drug effects. For most drugs on most endpoints, there was not a significant Sex × Dose interaction to suggest sex differences in drug effects. Moreover, in the four cases where a Sex × Dose interaction was observed, post hoc analysis revealed sex differences only at isolated doses without systematic differences in either potency or maximal effects. Overall, these findings suggest that the occasional statistical significance of sex differences was due more to random variation than to a reliable role of sex as a determinant of drug effects. Nonetheless, the post hoc power analysis of Sex × Dose interactions in Appendix A provides some guidance to sample sizes that would be required in future studies to investigate sex differences with these drugs in greater depth. 

## 4. Materials and Methods

### 4.1. Subjects

Male and female ICR mice (Envigo, Frederick, MD, USA) weighing 25–45 g were 6–8 weeks old upon arrival to the laboratory and were individually housed to prevent fighting with ad libitum access to water and food (Teklad LM-485 Mouse/Rat Diet; Envigo). Cages were mounted in racks in temperature-controlled rooms with a 12 h light/dark cycle (lights on at 6 a.m.) in a facility approved by the American Association for Accreditation of Laboratory Animal Care. All studies began at least 1 week after arrival at the laboratory and were usually completed during the second week after arrival. Animal-use protocols were approved by the Virginia Commonwealth University Institutional Animal Care and Use Committee and complied with the National Research Council Guide for the Care and Use of Laboratory Animals.

### 4.2. Drugs

Structures of the opioids tested in this study are shown in Figure 1. Hydrocodone bitartrate was provided by the National Institute on Drug Abuse Drug Supply Program (Rockville, MD, USA). Aprepitant was purchased from Cayman Chemical Co. (Ann Arbor, MI, USA). The KOR agonist nalfurafine and the mixed-action KOR/DOR agonists TK10 (NMF; *(2E)-N-[6α-17-(Cyclopropylmethyl)-3,14-dihydroxy-4,5α-epoxy-morphinan-6-yl]-3-(3-furanyl)-N-methyl-2-propenamide hydrochloride*), TK33 (*17-Cyclopropylmethyl-3,14β-dihydroxy-4,5α-epoxy-6α-[(2E)-3*′-*(furan-2*″-*yl)-N-methylprop-2-enamido]morphinan Hydrochloride*), and TK35 (*17-Cyclopropylmethyl-3,14β-dihydroxy-4,5α-epoxy-6β-[(2E)-3*′-*(furan-2*″-*yl)-N-methylprop-2-enamido]morphinan hydrochloride)* were provided by Dr. Zhang. Note that TK33 and TK35 are compounds #21 and 23, respectively, in a recent publication summarizing the pharmacology of these compounds in comparison to TK10 (NMF) and nalfurafine [15]. The DOR agonist SNC80 was provided by Dr. Kenner Rice. All drugs except the aprepitant were dissolved in sterile saline; the aprepitant was dissolved in 10% DMSO in saline. All drugs were administered SC in a volume of 10 mL/kg except the highest aprepitant dose, which was administered SC in a volume of 32 mL/kg due to limits in its solubility. Specifically, the highest concentration we could achieve in the designated vehicle was 1.0 mg/mL, and administration of this concentration in 32 mL/kg injection volume allowed us to administer a dose of 32 mg/kg. Lactic acid was diluted in sterile water and administered intraperitoneally (IP) in a volume of 10 mL/kg. 

### 4.3. Behavioral Testing

#### 4.3.1. General Experiment Design

Experimental procedures were identical to those described previously in studies that focused on MOR agonists [16,17]. Each mouse was tested only once, and with rare exceptions noted in the text, a different group of 12 mice (six females, six males) was used to test each dose of each drug. We have previously presented a detailed rationale for this group size and sex allocation [49]. For the present study, mice were generally tested in cohorts of up to 60 mice per week, which allowed testing of 5 different test drug doses administered alone or as a pretreatment to IP lactic acid in 5 different groups of 12 randomly allocated mice. For testing, groups of mice were brought from the housing room to the procedure room at least 1 h before injections, tested according to the procedures described below, returned to the housing room at the end of the day, and euthanized by the end of the week. Investigators were not blinded to treatment conditions because data collection for locomotor studies was automated by computer software and data from all mice were included for all experiments (i.e., there were no exclusion criteria, and no data were excluded). On test days for all procedures, mice were brought to the procedure room at least 1 h before drug injection and testing. 

#### 4.3.2. Apparatus

Locomotor activity was assessed as described previously using plexiglass and metal test boxes housed in sound-attenuating chambers (Med Associates, St. Albans, VT, USA) and located in a procedure room separate from the housing room. Each box had two adjacent compartments (16.8 × 12.7 cm^2^ floor area × 12.7 cm high) separated by a wall. One compartment had black walls with a bar floor. The other compartment had white walls with a wire-mesh floor. Additionally, each compartment had a clear plexiglass lid fitted with a house light as well as six photobeams arranged at 3 cm intervals across the long wall and 1 cm above the floor. Photobeam breaks were monitored by a microprocessor operating Med Associates software (MedPC 5). The wall separating the two compartments contained a central door (5 cm wide × 6 cm high) obstructed by a 1-inch (2.54 cm) tall stainless-steel wire-mesh barrier that had to be surmounted for mice to cross back and forth between the two compartments. 

#### 4.3.3. Procedure

Each drug was tested under two conditions. First, the vehicle and a range of drug doses were tested alone across a ≥1.5-log dose range in 0.5 log increments, with the vehicle or drug administered SC 30 min before a 15 min test session. The goal of these initial studies was to test a range from low doses that had no effect to either (1) doses that produced significant and >50% decreases in crosses and/or movement as an indicator of severe motor impairment as an unacceptable side effect, or (2) the upper limit of doses that could be administered given drug solubility. Second, the vehicle and a range of drug doses were tested as a pretreatment to IP injection of 0.56% lactic acid (0.56% IP lactic acid). The test drug or its vehicle was administered SC 30 min before a 15 min test session, and 0.56% lactic acid was administered IP 5 min before the session. The range of drug doses tested as a pretreatment to IP lactic acid was based on the results of the initial drug-alone study. Specifically, if a drug alone produced significant and ≥50% reduction in crosses and/or movement (as with most drugs in this study), then a range of lower doses with little or no motor-suppressing effect was tested in combination with IP lactic acid for possible antinociception. Additionally, because previous studies have indicated that many antinociceptive opioids produce inverted-U-shaped dose-effect curves under this procedure [16,17], we tested doses in smaller 0.25 log increments to optimize our potential for detecting antinociception if it were present. Alternatively, if a test drug failed to produce robust motor impairment up to the highest doses that could be tested (as with aprepitant in this study), then we used the same doses for drug-alone and drug + IP lactic acid studies.

At the start of each test session, the mouse was placed into the black compartment of the two-compartment chamber, and data were collected for 15 min. The dose ranges for each drug administered alone were as follows: hydrocodone (1–32 mg/kg), aprepitant (1–32 mg/kg), nalfurafine (0.0032–0.32 mg/kg), SNC80 (0.1–32 mg/kg), TK10 (0.01–0.32 mg/kg), TK33 (0.01–0.32 mg/kg), and TK35 (0.0032–0.1 mg/kg). Additionally, a 10:1 mixture of SNC80/nalfurafine was tested from doses of 0.032/0.0032 mg/kg SNC80/nalfurafine to 3.2/0.32 mg/kg SNC80/nalfurafine. The dose ranges for each test drug administered as a pretreatment to IP lactic acid were based on the effects of the test drug alone, as described above. 

#### 4.3.4. Dependent Measures

Two dependent measures were determined for each session in each mouse: (1) “Crosses”, defined as the number of crosses between the compartments and requiring mice to rear and surmount the vertical barrier in the doorway, and (2) “Movement”, defined as the total number of beam breaks summed across both compartments and requiring only horizontal locomotor activity. Data were also collected for “Bias,” defined as the proportion of each session spent on the side with black walls and the bar floor; however, as described previously [16,17], bias was not reliably altered by IP lactic acid as a pain-related behavioral effect, and as a result, drug effects on bias were not used to assess antinociception.

### 4.4. Data Analysis

Dose-effect data were averaged across mice within a given treatment and submitted to analysis that proceeded in two steps. First, data for a given manipulation were pooled across sexes and analyzed by one-way ANOVA. A significant ANOVA was followed by Dunnett’s post hoc test to compare test treatments with vehicle treatment. Second, data were segregated by sex and analyzed by two-way ANOVA, with sex as one of the variables. A significant main effect of Sex or Sex x Treatment interaction was followed by a Holm–Sidak post hoc test. Additionally, two-way ANOVA results were used to determine eta^2^ effect-size values for the Dose x Sex interaction using the equation: 

eta^2^ = sum of squares for the interaction/total sum of squares. These eta^2^ values were then used as we have described previously [16,17,49] for post hoc power analysis using G*Power 3 [50] to guide future studies that might focus on the role of sex as a determinant of drug effects. Specifically, G*Power was used to determine the Cohen’s F effect size, the current power (1-ß) of the analysis with the existing sample size, and the estimated sample size that would be required to detect a significant effect given the effect size and a criterion of power ≥ 0.8. 

In addition to this primary analysis of dose-effect data within a given drug, baseline data collected with control treatments were compared across drugs using *t*-tests or ANOVAs as appropriate. Prism 9.0 (GraphPad, La Jolla, CA, USA) was used for all ANOVAs and *t*-tests, and the criterion for significance was *p* < 0.05. 

## 5. Conclusions

The present results with hydrocodone and aprepitant further validated the sensitivity of this assay of pain-depressed behavior to clinically effective positive-control analgesics but not to negative-control non-analgesics that have produced false-positive effects in conventional assays of pain-stimulated behavior. In contrast to the hydrocodone effects, antinociception was not observed with the KOR agonist nalfurafine or DOR agonist SNC80 administered alone or in combination or with a series of nalfurafine analogs that have mixed KOR/DOR agonist activity. Taken together, these results suggest that caution is warranted in advancing KOR agonists, DOR agonists, or mixed-action KOR/DOR agonists as candidate analgesic drugs. Additionally, sex differences in drug effects were uncommon, and sex was not a reliable predictor of the potency or maximal effect of any drug.

## Figures and Tables

**Figure 1 molecules-29-03331-f001:**
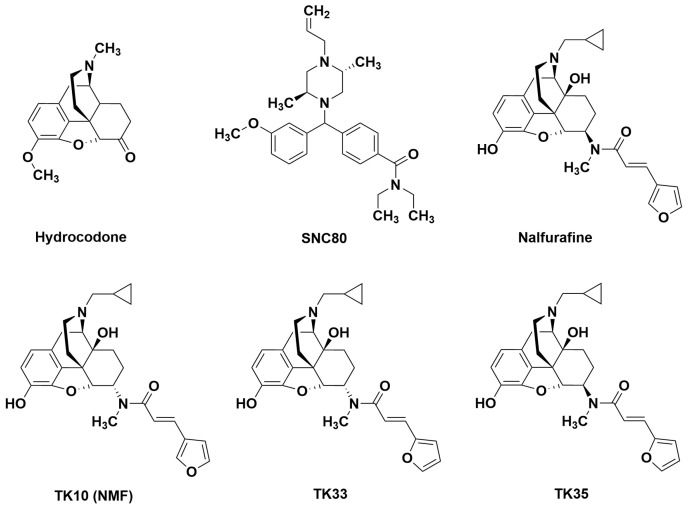
Structures of all opioid agonists evaluated in this study.

**Figure 2 molecules-29-03331-f002:**
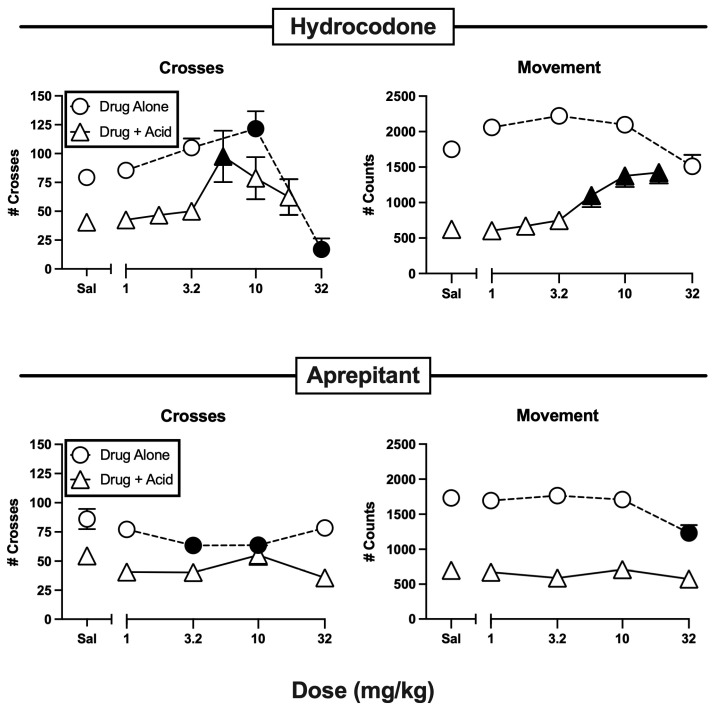
Antinociceptive effects of the positive-control opioid analgesic hydrocodone (**top panels**) but not the negative-control non-analgesic aprepitant (**bottom panels**) in an assay of pain-depressed behavior in mice. Abscissae: dose in mg/kg. Ordinates: number of crosses between compartments during each session (**left panels**; vertical locomotion) and number of movement counts during each session (**right panels**, horizontal locomotion). Each drug was tested both alone (circles) and as a pretreatment to IP 0.56% lactic acid as a noxious stimulus (triangles). Each point shows mean ± SEM in 12 mice, and filled points are significantly different from saline vehicle (Sal) as determined by a significant one-way ANOVA followed by Dunnett’s post hoc test (*p* < 0.05). The control data for saline vehicle alone and saline vehicle + 0.56% lactic acid over “Sal” in each panel were compared by *t*-test, and results are shown in Table 1.

**Figure 3 molecules-29-03331-f003:**
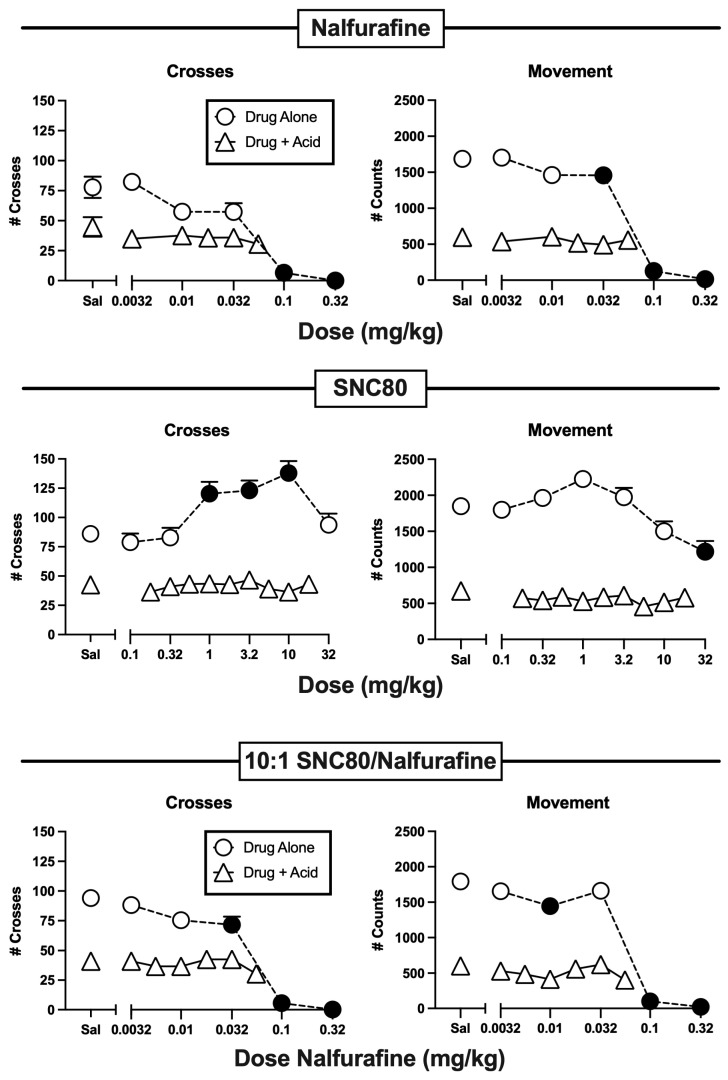
Lack of antinociceptive effects by the KOR agonist nalfurafine (**top panels**), DOR agonist SNC80 (**middle panels**), and a 10:1 SNC80/nalfurafine mixture (**bottom panels**). Other details as in Figure 2.

**Figure 4 molecules-29-03331-f004:**
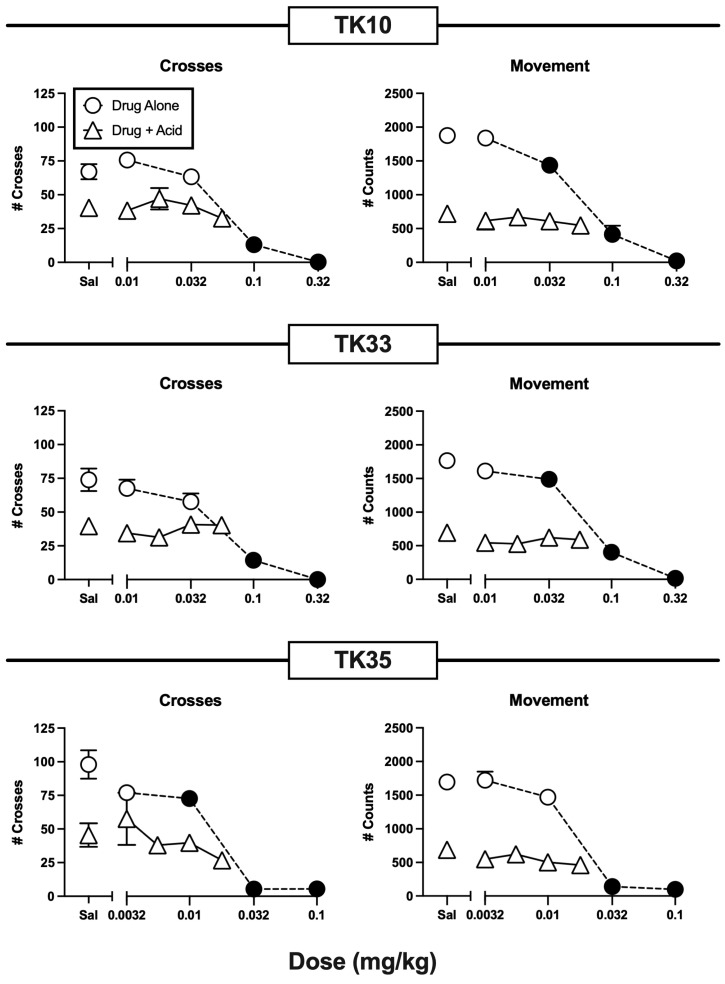
Lack of antinociceptive effects by the mixed-action KOR/DOR agonists TK10 (**top panels**), TK33 (**middle panels**), and TK35 (**bottom panels**). Other details as in Figure 2, except that the lowest dose of TK35 alone (0.0032 mg/kg) was tested in 6 mice rather than 12 mice.

**Table 1 molecules-29-03331-t001:** Under control conditions in the absence of drug treatment, IP lactic acid produced a consistent pain-related depression of both crosses and movement in each cohort of mice. T-test results are shown below for the comparison of data for saline vehicle alone and saline vehicle + 0.56% lactic acid (points above “Sal” in the figures). Drugs were evaluated for their effectiveness in alleviating this IP lactic acid-induced depression of behavior.

Test Drug	Crosses	Movement
Hydrocodone	t(22) = 4.76, *p* < 0.001	t(22) = 10.73, *p* < 0.001
Aprepitant	t(22) = 2.92, *p* = 0.0040	t(22) = 13.11, *p* = <0.001
Nalfurafine	t(22) = 2.73, *p* = 0.0061	t(22) = 11.16, *p* < 0.001
SNC80	t(22) = 6.52, *p* < 0.001	t(22) = 10.85, *p* < 0.001
10:1 SNC80/Nalfurafine	t(22) = 6.56, *p* < 0.001	t(22) = 12.08, *p* < 0.001
TK10	t(22) = 4.33, *p* = 0.0003	t(22) = 9.56, *p* < 0.001
TK33	t(22) = 3.60, *p* = 0.0008	t(22) = 10.54, *p* < 0.001
TK35	t(22) = 3.81, *p* = 0.0010	t(22) = 8.24, *p* < 0.001

**Table 2 molecules-29-03331-t002:** One-way ANOVA results for each drug to modify crosses and movement after administration of drug alone (circles in each figure) or drug + IP 0.56% lactic acid (triangles in each figure). Significant ANOVAs were followed by Dunnett’s post hoc test, and drug doses that produced effects significantly different from the vehicle are indicated by filled symbols in each panel. Each dose-effect curve was also evaluated by two-way ANOVA, with sex and dose as the two factors. Upper case letters indicate a significant Sex × Dose interaction, with higher scores in females (F) or males (M) for at least one dose. FM also indicates a significant Sex × Dose interaction but no difference between females and males at any dose in the post hoc test. Lower case letters indicate only a main effect of sex, with higher overall scores in females (f) or males (m).

Test Drug		Crosses	Movement
Hydrocodone	Alone	F(4,55) = 17.92, *p* < 0.0001-m	F(4,55) = 7.21, *p* < 0.0001
	+ IP lactic acid	F(6,77) = 2.55, *p* = 0.0264	F(6,77) = 9.08, *p* < 0.0001
Aprepitant	Alone	F(4,55) = 2.77, *p* = 0.0359-F	F(4,55) = 8.16, *p* < 0.0001
	+ IP lactic acid	F(4,55) = 1.75, *p* = 0.1525-m	F(4,55) = 0.88, *p* = 0.4846
Nalfurafine	Alone	F(5,66) = 34.34, *p* < 0.0001-m	F(5,66) = 156.60, *p* < 0.0001
	+ IP lactic acid	F(5,66) = 0.88, *p* = 0.4991-m	F(5,66) = 0.51, *p* = 0.7679-f
SNC80	Alone	F(6,77) = 7.34, *p* < 0.0001	F(6,77) = 8.38, *p* < 0.0001-f
	+ IP lactic acid	F(9,110) = 0.45, *p* < 0.9061-FM	F(9,110) = 1.25, *p* = 0.2722-F
10:1 SNC/Nal	Alone	F(5,66) = 56.84, *p* < 0.0001	F(5,66) = 123.00, *p* < 0.0001
	+ IP lactic acid	F(6,77) = 1.19, *p* = 0.32	F(6,77) = 1.63, *p* = 0.1513
TK10	Alone	F(4,55) = 62.16, *p* < 0.0001	F(4,55) = 89.24, *p* < 0.0001-f
	+ IP lactic acid	F(4,55) = 0.94, *p* = 0.4480	F(4,55) = 0.41, *p* = 0.8002
TK33	Alone	F(4,55) = 33.38, *p* < 0.0001	F(4,55) = 110.80, *p* < 0.0001
	+ IP lactic acid	F(4,55) = 1.00, *p* = 0.4142-m	F(4,55) = 0.88, *p* = 0.4848
TK35	Alone	F(4,49) = 43.57, *p* < 0.0001	F(4,49) = 93.17, *p* < 0.0001-f
	+ IP lactic acid	F(4,55) = 1.22, *p* = 0.3144	(4,55) = 1.81, *p* = 0.1398-F

## Data Availability

This manuscript contains data from locomotor behavioral studies in mice. The authors declare that all mean ± error behavioral data are available within the paper and its Appendix A, and raw data are available by request to the corresponding author.

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
