# Peer review of "Effects of Selective and Mixed-Action Kappa and Delta Opioid Receptor Agonists on Pain-Related Behavioral Depression in Mice"

_molecules, 2024, doi:10.3390/molecules29143331_

Round 1

Reviewer 1 Report

Comments and Suggestions for Authors

In the manuscript titled “Effects of Selective and Mixed-Action Kappa and Delta Opioid Receptor Agonists on Pain-Related Behavioral Depression in Mice” by Negus et al., the authors have reported the use of intraperitoneal acid-induced behavioural depression study to compare the analgesic effect of selective (nalfuratine, SNC80) and mixed-action kappa and delta opioid receptor agonists (TK10, TK33, TK35) with a positive control analgesic (hydrocodone) and negative control analgesic (aprepitant). Based on the findings, the authors concluded that kappa and delta opioid receptor agonists are potentially risky to be developed as candidate analgesic drugs.

The manuscript generally reads well, however, there are a few comments which require the authors’ attention as below:-

Page 3, line 114 – The authors stated that there are significant increases for both crosses and movement when hydrocodone was administered alone at low doses (3.2 or 10mg/kg), however, in Figure 2, no significant increase was observed for the movement across all doses tested.

Both Figures 3 and 4 illustrated that the doses tested when the drugs were given alone and in the presence of intraperitoneal lactic acid were not comparable. In other words, high doses of nalfurafine, TK10, TK33 and TK35 were not tested in the presence of lactic acid. Any reason for the inconsistency in the selection of doses for testing between drug alone and in the presence of lactic acid? In Figure 2, it is shown that when hydrocodone was given alone at higher doses (>10mg/kg), there was a drop in movement and crosses, however, in the presence of lactic acid, the trend was reversed. Would the same trend be observed if the same (higher doses) were tested in the presence of lactic acid for the four tested drugs? Also, what could be the explanation for this observation (that higher doses negatively disrupt the motor function at high doses, and yet it increases movement in the presence of lactic acid)?

Page 4, line 195 – Supplemental Table 1 is not provided for review.

Page 11, line 411 – The authors cautioned that KOR agonist, DOR agonist or mixed KOR/DOR agonist are potentially not good analgesic candidates based on the observations that these agents were not able to reverse locomotor depression induced by IP lactic acid. However, this is quite misleading as this assay only detects locomotor depression, not the analgesic effect. In other words, how can we rule out the possibility that kappa agonists can both depress locomotor as well as provide analgesic effects at the same time? In my opinion, to claim that kappa agonist or delta agonist is not a good analgesic candidate based on just locomotor depression reversal is an overstatement.

Reviewer 2 Report

Comments and Suggestions for Authors

The manuscript by Negus and colleagues seeks to examine the effects of selective and mixed-action kappa and delta opioid receptor agonists on pain depressed mouse behaviors. The manuscript is of high reviewer interest, is the experiments are methodically and logically performed, and is timely given the need of novel pain therapeutics. Nonetheless, this manuscript could be improved with the below suggestions taken into consideration.

Introduction

1.      “These procedures have improved preclinical-to-clinical translational validity because clinically effective analgesics are effective, whereas drugs that produce motor impairment only exacerbate pain-related behavioral depression and do not produce false-positive effects.” This sentence could be more clearly worded. As written, the sentence lends to a bit of a strawman argument that clinically effective drugs are effective preclinically and does not really convey the point that this reviewer is pretty sure the authors intend to make here.

Results

2.      “ … as a pretreatment to IP 0.56%  acid.” Here and elsewhere throughout the manuscript the authors need to distinguish that lactic acid is used. This reviewer understands that this is likely shorthand writing by the authors, but to a naïve reader, this point might be confusing, as other types of ‘acid’ can be used in preclinical pain models.

3.      For the data analyzed in Table 1, I believe that this data is the comparison of each drug group’s veh + veh and acid + veh behavior, where there are no drugs administered to the animal. As worded, this is a bit difficult to discern, and took this reviewer reading through this section several times to grasp what the treatment conditions were. The table legend/ section for this data could be more clearly worded to avoid potential confusion.

Text – Supplementary Materials

4.      “Supplementary Materials: To be determined.” This should be determined by the authors and if needed, included.

Reviewer 3 Report

Comments and Suggestions for Authors

Dear Authors,

The article I was given to review is an extremely interesting and a large-scale study of the potential analgesic effects of newly-synthesized substances on an induced pain rodent model. The modern approach to the search and discovery of new drugs is based on modifying known molecules. The research conducted has the potential to benefit the elucidation of mechanisms and potential targets for influencing pain, including improving the extrapolation of results from experimental animals to humans.

However, I believe that the article could be published after major revision.

Here are my recommendations, as well as some inaccuracies and omissions, as follows.

1) The sentences in the introductory part of the article are too long; I recommend that they separate. Moreover, most of them are formed by using the punctuation mark ";".

2) The paragraph from lines 51 to 68 is more appropriate to be included in the discussion part.

3) I have a question for the authors regarding the chosen pain induction methodology. Lactic acid, as well as acetic acid, injected IP cause pain of visceral origin, which is successfully antagonized by the administration of nonsteroidal anti-inflammatory drugs and other analgesics with a peripheral mechanism of action. In their experiment, Garner et al. (2021) (doi: 10.3389/fpain.2021.673940) after the use of lactic acid IP, an analgesic effect was exhibited by ketoprofen, while morphine was effective only when the mice were injected with a highly diluted lactic acid solution. The conclusions of Stevenson et al. (2009) (doi: 10.1016/j.lfs.2009.06.006) are also in this direction: "Morphine blocked both acid-induced stimulation of writhing and depression of locomotion, although it was 56-fold less potent in the assay of acid-depressed locomotion." Why did the authors choose this pain model when other central pain models, such as the hot plate test, are used to monitor the analgesic potential of opioid receptor agonists? Accordingly, morphine is then an appropriate referent.

4) More impressions about the methodological part of the manuscript. Getting acquainted with the materials provided to me for review, the following is striking:

-          the description of the experimental protocol is rather vague and does not comply with the requirement for an accurate detailed description and reproducibility of the conducted research. The writing overflows with references to other developments in which the protocol has been introduced and verified, but there is no concrete data here. For example, only in the results section (lines 85-86) does it become clear that the experimental mice were studied over a period of 6 months. Also, the total number of experienced rodents and their distribution by gender is not specified.

-          Another omission is the grouping – how many of the animals were treated with each substance of the respective dose.

-          The methods section also does not specify the doses applied or the reason for their selection.

-          The reason for individual housing of mice during the experiments was not stated. Usually the rodents are grouped in a minimum of five animals in a cage; otherwise, it is considered a social isolation protocol for inducing aggression and other behavior changes.

-          Please explain the reason for administering a larger volume of aprepitant solution compared to all other substances administered in a standard volume for IP injection in mice, namely 0.1 ml / 10 g bw.

-          The sentences in lines 345-356 are confusing. I recommend that the authors include a scheme of the experimental design so that the reader of the article can better navigate the long-term experiment they are conducting.

-          I recommend that the description of the apparatus include who developed and validated it for monitoring behavior in mice with the indicated model of induced pain (4.3.2. Apparatus)

-          In section 4.3.3. the authors indicate the dose ranges of the administered substances, but do not describe how many doses from the range of each substance they were selected from. The reader of the article can be guided only by the presented figures with results

5) Given the enormous amount of work that has been done, with a complex and lengthy protocol, the results are rather limited in presentation.

-          The results of Figures 2, 3 and 4 do not show the dynamics of behavior in the two sexes of the experimental animals. These data are presented in table 2, but the doses are missing there. These inaccuracies make interpretation of the results difficult.

-          There is a discrepancy in the titles of the two scoreboards. Table 1 compares the effects of "Vehicle Alone and Vehicle + IP 0.56% acid", and in Table 2 - "drug alone (circles in each figure) or drug + IP 0.56% acid". This is confusing because of the different meaning of a drug vehicle and the drug itself...

-          Please specify "lactic acid" instead of just "acid"

-          Also, the font size in the two tables is different.

-          Why are explanations for the data presented in the figures included in the table titles?

-          Additional materials are mentioned in the article (lines 195 and 198), but they were not made available for review. Additionally, the article template in the Supplementary Materials section states "To be determined."

6) The conclusion part should be extended with illations about the different doses that were administered, the dose/response relationships that were found, and the behavioral differences between the two sexes that were found.

Review date

June 26th 2024

Round 2

Reviewer 3 Report

Comments and Suggestions for Authors

Dear authors,
The corrections to the manuscript are found to be satisfactory, as well as the reasons provided for not addressing some comments.

Author Response

We thank the reviewer for their kind comments regarding the paper, and we have addressed both comments as described below.  I apologize for the typos!

1. We have added text to the Figure 2 legend to identify points that were compared by t-test for results shown in Table 1.
2. We have corrected the typos in sentence 322-326.